# The Prostacyclin Analogue Iloprost Modulates CXCL10 in Systemic Sclerosis

**DOI:** 10.3390/ijms231710150

**Published:** 2022-09-05

**Authors:** Tania Colasanti, Katia Stefanantoni, Cristina Fantini, Clarissa Corinaldesi, Massimiliano Vasile, Francesco Marampon, Luigi Di Luigi, Cristina Antinozzi, Paolo Sgrò, Andrea Lenzi, Valeria Riccieri, Clara Crescioli

**Affiliations:** 1Rheumatology Unit, Department of Clinical Internal, Anesthesiological and Cardiovascular Sciences, Sapienza University of Rome, Viale del Policlinico, 155-00161 Rome, Italy; 2Unit of Biology and Genetics of Movement, Department of Movement, Human and Health Sciences, University of Rome “Foro Italico”, Piazza Lauro de Bosis, 006-00135 Rome, Italy; 3Unit of Endocrinology, Department of Movement, Human and Health Sciences, University of Rome “Foro Italico”, Piazza Lauro de Bosis, 006-00135 Rome, Italy; 4Institute for Cancer Genetics, Columbia University, New York, NY 10027, USA; 5Department of Radiotherapy, Sapienza University of Rome, Viale del Policlinico, 155-00161 Rome, Italy; 6Department of Experimental Medicine, Sapienza University of Rome, Viale del Policlinico, 155-00161 Rome, Italy

**Keywords:** iloprost, human endothelial cells, human fibroblasts, CXCL10, systemic scleroderma

## Abstract

The prostacyclin analogue iloprost is used to treat vascular alterations and digital ulcers, the early derangements manifesting in systemic sclerosis (SSc), an autoimmune disease leading to skin and organ fibrosis. Bioindicator(s) of SSc onset and progress are still lacking and the therapeutic approach remains a challenge. The T helper 1 (Th1) chemokine interferon (IFN)γ-induced protein 10 (IP-10/CXCL10) associates with disease progression and worse prognosis. Endothelial cells and fibroblasts, under Th1-dominance, release CXCL10, further enhancing SSc’s detrimental status. We analyzed the effect of iloprost on CXCL10 in endothelial cells, dermal fibroblasts, and in the serum of SSc patients. Human endothelial cells and dermal fibroblasts activated with IFNγ/Tumor Necrosis Factor (TNF)α, with/without iloprost, were investigated for CXCL10 secretion/expression and for intracellular signaling cascade underlying chemokine release (Signal Transducer and Activator of Transcription 1, STAT1; Nuclear Factor kappa-light-chain-enhancer of activated B cells, NF-kB; c-Jun NH_2_-terminal kinase, JNK: Phosphatidyl-Inositol 3-kinase (PI3K)/protein kinase B, AKT; Extracellular signal-Regulated Kinase 1/2, ERK1/2). CXCL10 was quantified in sera from 25 patients taking iloprost, satisfying the American College of Rheumatology (ACR)/European Alliance of Associations for Rheumatology (EULAR) 2013 classification criteria for SSc, and in sera from 20 SSc sex/age-matched subjects without therapy, previously collected. In human endothelial cells and fibroblasts, iloprost targeted CXCL10, almost preventing IFNγ/TNFα-dependent cascade activation in endothelial cells. In SSc subjects taking iloprost, serum CXCL10 was lower. These in vitro and in vivo data suggest a potential role of iloprost to limit CXCL10 at local vascular/dermal and systemic levels in SSc and warrant further translational research aimed to ameliorate SSc understanding/management.

## 1. Introduction

Systemic sclerosis (SSc) is an autoimmune disease defined as a “vascular” illness involving severe endothelial derangements, which promote deep alteration in skin and organ architecture, ending in multi-organ fibrosis due to collagen overproduction [1,2]. In SSc, fibrosis extension inversely associates with survival, which declines to 39%, in presence of skin or internal organ (gastrointestinal tract, lung, kidney, heart) fibrosis, from 72%, in presence of no/mild organ involvement [3]. Vascular alteration and fibroproliferative vasculopathy act as early triggers in SSc pathogenesis [4,5].

Indeed, the deregulation of vascular tone control, which is clinically detectable as the Raynaud phenomenon (RP), often ending in digital ulcers (DU), occurs early in SSc and may precede, by months or even years, organ involvement and damage [6,7,8]. In particular, the interaction between the altered endothelium and skin, organ, or tissues occurs mainly through the release of some immunoactive/inflammatory molecules, especially T helper 1 (Th1) type cytokines and chemokines [9,10]. Currently, there is no bioindicator of these initial processes. Hence, biomolecules and paths actively participating and engaging in early vascular deregulation can represent potential biomarkers and therapeutic targets.

Among the involved biomediators, the IFNγ-induced 10 kDa protein (IP-10/CXCL10) is a Th1 type small chemokine playing a dual role, in inflammation and vascular assessment [11,12]. This chemokine seems to be highly involved in SSc. Indeed, higher CXCL10 is not only associated with a more severe disease prognosis [9,13,14] but, remarkably, precedes the development of definite SSc, from a preclinical condition of undifferentiated connective tissue disease (UCTD) at risk for SSc, so far referred to as very early diagnosis of systemic sclerosis (VEDOSS) [14,15,16]. Normally, this chemokine is produced by immunocytes [17,18]. Noticeably, upon the Th1 challenge, CXCL10 can be released by some other human cell types, i.e., endothelial cells, renal cells, striated cells, and fibroblasts, potentially representing a biotarget within some defined disease-related contexts [19,20,21,22,23,24,25].

Currently, the prostacyclin analogue iloprost is included in the first-choice medications in SSc for the treatment of severe RP, DU, and pulmonary arterial hypertension [26]. Iloprost acts by binding mainly on the prostacyclin receptor (IP receptor), causing the elevation of cyclic AMP (cAMP) [27], which can play a regulation (decrease) of CXCL10 levels [28,29].

This study aims in vitro to assess whether and how iloprost can target CXCL10 release by human endothelial cells and fibroblasts. For this purpose, we analyzed CXCL10 protein secretion and gene expression by human endothelial cells and dermal fibroblasts maintained in a Th1-dominant microenvironment with and without iloprost, along with some intracellular paths involved in chemokine release. The in vivo part of the study is to evaluate circulating CXCL10 levels in SSc subjects taking iloprost, either alone or in addition to other treatments (disease-modifying anti-rheumatic drugs, DMARDs and corticosteroids, CCs), and in a group of SSc subjects without therapy previously analyzed [14].

## 2. Results

### 2.1. Iloprost Decreased Interferon (IFN)γ and Tumor Necrosis Factor (TNF)α-Induced CXCL10 Secretion in Human Endothelial Cells and Dermal Fibroblasts

In Human fetal aortic endothelial cells (Hfaec) and Human dermal fibroblasts (Hdf), iloprost significantly decreased CXCL10 protein secretion induced by IFNγ and TNFα (1003.5 ± 254.4 pg/µg vs. 1720 ± 206.5 pg/µg, *p* < 0.05 and 721.8 ± 219.2 pg/µg vs. 1387 ± 189.9 pg/µg, *p* < 0.05, respectively) (Figure 1A,B). CXCL10 release was virtually absent in both endothelial and fibroblast controls (*p* < 0.001 vs. IFNγ and TNFα-induced secretion). At variance with protein release, the specific mRNA expression decreased with iloprost in Hfaec, while in Hdf seemed to increase with respect to IFNγ and TNFα cytokine-induced expression (*p* < 0.01) (inset of Figure 1A,B, respectively).

### 2.2. Iloprost Targeted IFNγ and TNFα-Dependent Signaling Only in Human Endothelial Cells

In Hfaec, iloprost virtually prevented IFNγ and TNFα-induced activation (taken as 100%) of Signal Transducer and Activator of Transcription 1 (STAT1; % of inhibition: 93.1 ± 1.2%, *p* < 0.01), Nuclear Factor kappa-light-chain-enhancer of activated B cells (NF-kB; % of inhibition: 92 ± 3.3%, *p* < 0.01), and significantly decreased Extracellular signal-Regulated Kinase (ERK)1/2 (% of inhibition: 26.9 ± 4.4%, *p* < 0.05) (Figure 2A,B,E), without affecting c-Jun NH_2_-terminal kinase (JNK) and Phosphatidyl-Inositol 3-kinase (PI3K)/protein kinase B (AKT) (Figure 2C,D).

Conversely, in Hdf iloprost almost did not affect IFNγ and TNFα-dependent cascade (Figure 2F–H,J), targeting only AKT (% of inhibition: 30.4 ± 17.8%, *p* < 0.05 vs. cytokine-induced phosphorylation, taken as 100%) (Figure 2I).

### 2.3. Different Intracellular Path Engagement in CXCL10 Protein Release by Human Endothelial Cells and Fibroblasts

Assays with chemical-specific path inhibitors showed that NF-kB and ERK1/2 were the most involved intracellular pathways underlying CXCL10 secretion in Hfaec and Hdf, respectively (Figure 3A,B). Indeed, the blockage of NF-kB induced by BAY11-7082 in Hfaec and the blockage of ERK1/2 with U0126 in Hdf almost nullified CXCL10 release (% of inhibition: 94.1 ± 5.9% and 95.1 ± 2.7%, respectively, *p* < 0.001 vs. IFNγ and TNFα-induced secretion, taken as 100%).

In Hfaec, chemokine release was significantly reduced also blocking JNK with SP600125 (% of inhibition: 77.6 ± 12.2%, *p* < 0.01 vs. IFNγ and TNFα-induced secretion) and ERK1/2 (% of inhibition: 64.4 ± 12.6%, *p* < 0.01 vs. IFNγ and TNFα-induced secretion).

In Hdf, IFNγ and TNFα-induced CXCL10 secretion was also significantly decreased by NF-kB or JNK blockage (% of inhibition: 83.3 ± 3.4% and 74.7 ± 6.6%, *p* < 0.001) and, to a lesser extent, by STAT1 blockage with fludarabine (% of inhibition: 46.8 ± 14.8%, *p* < 0.05).

### 2.4. Iloprost Reduced Serum CXCL10 in SSc

From in vivo investigation, we observed a significantly lower serum CXCL10 in subjects taking iloprost, either alone (260.3 ± 45.3 pg/mL, *p* < 0.001) or in combination with disease-modifying anti-rheumatic drugs (DMARDs) or corticosteroids (CCs) (552.3 ± 198.3 pg/mL and 407.4 ± 76.5 pg/mL, respectively, *p* < 0.01), as compared with SSc subjects without therapy (806.3 ± 110.6 pg/mL) (Figure 4).

## 3. Discussion

The main finding of this study is that iloprost inhibits CXCL10 secretion in activated human endothelial cells and dermal fibroblasts, almost preventing the activation of the paths synergistically involved in chemokine release, such as STAT1 and NF-kB, in endothelial cells. SSc subjects taking iloprost show significantly lower levels of circulating CXCL10, compared with a group of SSc subjects without therapy.

In SSc, vascular rearrangement occurs as a very early event, when the diagnosis remains very difficult [30]. Vascular damage involvement, clinically manifest with RP and DU, plays a crucial role in SSc pathogenesis [4], leading to skin and tissue fibrosis and, consequently, organ dysfunction and failure in later stages [31]. Thus, the challenge in translational research is to find early vascular biomarker(s)/potential pharmacological target(s), mirroring pathological process onset and progression, to be useful in disease prevention, prognosis, or treatment [32].

CXC motif chemokines are well-known factors widely contributing to vascular injury and damage [33]. These chemokines have been shown to attract and coordinate the migration of immunocytes positive for their specific receptors, such as CXCR^+^ Th1 cells, to mediate angiostatic activity and direct vascular damaging effects [34]. Among CXC chemokines, CXCL10 is known to contribute to SSc-associated vasculopathy, through the interaction with the specific receptor subtype [35,36,37,38]. In particular, SSc patients with higher CXCL10 serum levels show a more severe prognosis phenotype [9,39]. This aspect is associated with the capillaroscopic pattern worsening and disease progression from UCTD at risk for SSc, so far referred to as VEDOSS, to definite SSc disease [14,16]. CXCL10 is one of the major triggers of early Th1/Th17-driven activation and polarization in allo- and autoimmunity, establishing and perpetuating a self-detrimental loop between local and systemic levels from the earliest stages of the response [14,19,37,39,40,41]. CXCL10 local production induced by infiltrating immunocytes releasing cytokines at inflammatory sites—such as IFNγ and TNFα—recalls more Th1 cells positive for CXCR3 (the specific chemokine receptor). These cells, in turn, further enhance CXCL10 local release [19,35,36,37]. Thus far, the control of these early processes aimed at limiting the CXCL10 vicious loop, might be helpful in the therapeutic approach. The release of CXCL10 is synergistically promoted by IFNγ and TNFα, involving the activation of the transcription factors STAT1 and NF-kB, the prototypic paths of IFNγ and TNFα, respectively, as we previously reported in different human cell types, such as renal tubular cells, endothelial cells, cardiomyocytes, and skeletal muscle cells [19,20,22,23,37,39,42]. Herein, we document that iloprost almost prevents IFNγ/TNFα-induced STAT1 and NF-kB activation in human endothelial cells, as shown by the virtual absence of the phosphorylation, in association with a significant reduction (by about 50%) of cytokine-promoted CXCL10 secretion. These nuclear factors are maintained in the cytoplasm in their inactive form and, upon phosphorylation, translocate into the nucleus inducing, among other effects, chemokine/cytokine upregulation [43]. Furthermore, iloprost significantly counteracts the phosphorylation of ERK1/2, a signaling path involved in CXCL10 protein secretion together with NF-kB and JNK, as shown by the experiments with path inhibitors performed in this study. In this experimental assay, the lack of significant data on STAT1 inhibition may depend on the high variability of obtained numbers due, in turn, to some possible toxic effect exerted by the drug onto human endothelial cells [44]. These observations and the absence of any effect on CXCL10 specific mRNA expression by iloprost suggest a regulation at a post-transcriptional level in Hfaec. We could speculate that this modulation adds up to the effects mediated by iloprost-induced regulation of cAMP level, known to improve endothelial inflammatory status. Indeed, iloprost binds IP receptor, which is coupled via Gs to adenylyl cyclase. The consequent increase in cAMP production, in turn, can counteract inflammatory activity mediated by critical factors, such as NF-kB [45]. A similar effect is described upon iloprost binding with the peroxisome proliferator-activated receptor γ (PPARγ), which suppresses NF-kB signaling [46].

In Hdf, iloprost seemed not able to modify the tested signal transduction cascade, except for AKT phosphorylation. Healthy human fibroblasts challenged by IFNγ/TNFα secreted CXCL10, virtually absent in control cells, in association with activation of STAT1, NF-kB, and AKT, as previously and herein reported [23]. Since AKT activation is known to be largely involved in CXCL10 secretion by many cell types in different disease conditions [47,48,49], in human fibroblasts it is likely that iloprost-induced AKT impairment could be enough to reduce CXCL10 protein release. Concerning the discrepancy observed between chemokine protein and gene expression, we could speculate that iloprost-induced CXCL10 mRNA increase is likely due to mRNA accumulation, rather than an increased transcription, since it is not mirrored by a rise in protein level. Our ongoing and future studies will be of help to verify this hypothesis.

Previous investigations reported the involvement of STAT3, Focal adhesion kinase (FAK), Glycogen synthase kinase (GSK)3αβ, and Protein kinase (PK)Cδ signaling paths in CXCL10 upregulation [50,51]. From our experimental data obtained with signaling blockers, JNK, NF-kB, and ERK1/2, although not targeted by the drug, seemed to be the main paths involved in CXCL10 release from Hdf. Some other experiments are mandatory to further verify the mechanistic aspects of the iloprost effect on human fibroblasts.

Concerning in vivo data, we observed that SSc subjects taking iloprost, either alone or combined with DMARDs and CCs, showed significantly lower circulating levels, as compared with the population of SSc subjects previously studied and herein re-analyzed for comparison. This might be intriguing. Indeed, we could speculate that lowering serum CXCL10 might be of particular interest, considering that higher circulating CXCL10 is associated with a shift from UCTD at risk for SSc, so far referred to as VEDOSS, to defined SSc, as previously addressed [14,16]. Further in vivo studies in a higher number of SSc subjects before and after the treatment with iloprost, combined or not with other therapies, are mandatory.

Furthermore, from in vivo data, we could speculate that the lower serum CXCL10 may mirror the ability of iloprost to counteract CXCL10 release from local tissues, differently from DMARDs and CCs, which are known to mainly target immune cells.

The ability to downregulate chemokine secretion directly in endothelial tissue might be of high interest in the early therapeutic approach, since vascular injury is of primary importance in the pathogenesis of SSc, from the very early onset of the disease [52].

## 4. Materials and Methods

### 4.1. Subjects

Sera from 25 patients admitted to the Rheumatology Unit of the Sapienza University of Rome with a diagnosis of SSc, according to the American College of Rheumatology (ACR)/European Alliance of Associations for Rheumatology (EULAR) 2013 classification criteria [53] were collected. Patients’ demographic, clinical and therapeutic data were recorded at the baseline visit and reported in Table 1. SSc patients were in treatment with iloprost alone (7), iloprost and DMARDs (9), or iloprost and CCs (9). For comparison, we reported CXCL10 values in sera from a cohort of sex- and age-matched SSc subjects (20) before starting any therapy [14]. The study protocol was approved by the Local Ethics Committee (Prot. n° 416/11). The procedures involving human participants were in accordance with the Declaration of Helsinki and informed consent was obtained from all the participants in the study. All blood samples were collected from the peripheral vein; serum aliquots obtained by centrifugation (3000 rpm for 10 min at 4 °C) were stored at −80 °C until the analysis.

### 4.2. Chemicals

Endothelial Growth Medium (EGM) for Hfaec was from LONZA (Basel, Switzerland). Medium 106 with the addition of Low Serum Growth Supplement for Hdf was purchased from ThermoFisher Scientific (Waltham, MA, USA). Fetal bovine serum (FBS) was from Cytiva HyClone (Sigma Aldrich Corp., St. Louis, MO, USA). Recombinant human IFNγ and TNFα were purchased from Peprotech (Rocky Hill, NJ, USA). Plastic wares for cell cultures and disposable filtration units for growth media handling were purchased from Corning Inc. (Corning, NY, USA). 2-mercaptoethanol and SYBR Green Polymerase Chain Reaction (PCR) Master Mix for quantitative PCR (qPCR) were from Life Technologies, Inc. Laboratories (Carlsbad, CA, USA). Ca^2+^/Mg^2+^-free phosphate-buffered saline (PBS), bovine serum albumin (BSA) fraction V, penicillin/streptomycin antibiotics, collagenase type IV, EDTA-trypsin solution, NaOH, Tris-buffered saline (TBS), STAT1 inhibitor fludarabine, NF-kB inhibitor BAY11-7082, JNK inhibitor SP600125, AKT inhibitor LY294002, ERK inhibitor U0126, rabbit IgG anti-β-actin and synthetic prostacyclin analogue iloprost were purchased from Sigma Aldrich Corp. (St. Louis, MO, USA). The antibodies for western blot analysis anti-phospho-JNK (p-JNK) (G-7, sc-6254), anti-JNK (D-2, sc-7345), anti-phospho-STAT1 (p-STAT1) (A-2, sc-8394), anti-STAT1 (B-9, sc-271661), anti-phospho-NF-kB (p-NF-kB) (27.Ser 536, sc-136548), anti-NF-kB p50 (E-10, sc-8414), anti-phospho-ERK1/2 (p-ERK1/2) (pT202/pY204.22A, sc-136521), anti-ERK1/2 (C-9, sc-514302), anti-phospho-AKT1/2/3 (p-AKT1/2/3) (C-11, sc-514032), anti-AKT1/2/3 (B-1, sc-5298), and horseradish peroxidase (HRP)-conjugate anti-mouse and anti-rabbit IgG were all from Santa Cruz Biotechnology, Inc. (Dallas, TX, USA).

### 4.3. Cell Cultures

Human fetal aortic endothelial cells (Hfaec) were obtained from fetal tissues collected after voluntary abortion (10–12 weeks of gestation), characterized, and maintained as previously described [19,54]. Briefly, Hfaec were isolated from four fetal aortic ascendant tracts. Tissues were washed with PBS and cut longitudinally to gently scrape cells with a scalpel from the internal side, in presence of 1 mg/mL bacterial collagenase type IV. Cells were then washed from the scalpel, collected, and cultured in EGM supplemented with inactivated 10% FBS, 100 U/mL penicillin, and 100 μg/mL streptomycin. Cells expressed von Willebrand factor (vWF) and specific mRNA of Tie-2, both established markers for endothelial cells [55,56]. Confluent cells were split in 1:2 ratio with EDTA-trypsin solution (0.2–0.5%) and used within the 5th passage.

Legal abortions were performed in an authorized hospital, and written consents for human fetal tissues to be stored and used for research were obtained. The use of human fetal tissue for research purposes conforms with the principles outlined in the Declaration of Helsinki and was approved by the Committee for investigation in humans of the Azienda Ospedaliero-Universitaria Careggi, Florence, Italy (Prot. n^◦^ 6783-04).

Primary Human dermal fibroblasts (Hdf), isolated from adult skin (ThermoFisher Scientific, Waltham, MA, USA), were cultured in Medium 106 with the addition of Low Serum Growth Supplement (ThermoFisher Scientific, Waltham, MA, USA), following manufacturer’s instructions, and with 100 U/mL penicillin and 100 μg/mL streptomycin.

Confluent cells were split in 1:2 ratio with EDTA-trypsin solution (0.2–0.5%) and used within the 4th passage, after cell thawing.

Both cell types were cultured at 37 °C, in a fully humidified atmosphere of 95% air and 5% CO_2_.

For the CXCL10 secretion assay, 4000 cells/well were seeded onto 96-well flat-bottom plates and maintained for 24 h in the growth medium at 5% CO_2_ and 37 °C. After overnight starvation in serum- and phenol red-free medium, cells were stimulated for 24 h with IFNγ (1000 U/mL) and TNFα (10 ng/mL), in the presence or absence of 30 min pretreatment with iloprost (8 nM) or 1 h pretreatment with the path selective inhibitors fludarabine (50 μM, for STAT1), BAY11-7082 (20 μM, for NF-kB), SP600125 (100 μM, for JNK), LY294002 (15 μM, for PI3K/AKT), U0126 (20 μM, for ERK1/2).

Cells in serum-free medium containing 0.1% BSA and vehicle (absolute ethanol 0.47% v/v) were used as control. Iloprost concentration selected was maintained within the near-therapeutic range, according to the pharmacokinetic parameters (maximum drug concentration, Cmax, and area under the time-concentration curve, AUC).

For mRNA analysis, 50,000 cells/well were seeded and maintained as previously reported [21]. After 12 h starvation, cells were stimulated for 24 h with IFNγ (1000 U/mL) and TNFα (10 ng/mL), with/without pretreatment with iloprost (8 nM) for 30 min in serum-free medium containing 0.1% BSA, and then analyzed [57].

For Western blot analysis, Hfaec and Hdf were exposed for 10 min to IFNγ (1000 U/mL) and TNFα (10 ng/mL), with/without 30 min pre-incubation with iloprost (8 nM). Cells maintained under the same conditions with vehicle (absolute ethanol 0.47% v/v)/without the drug were used as control.

### 4.4. Enzyme-Linked ImmunoSorbent Assay (ELISA)

Supernatants from both Hfaec and Hdf were harvested, centrifuged at 4 °C, 10 min, aliquoted, and stored at −20 °C until the assays. CXCL10 was determined in cell supernatants and sera, using commercially available kits (Quantikine ELISA Human CXCL10/IP-10 Immunoassay, R&D Systems Inc., Minneapolis, MN, USA), according to the manufacturer’s recommendations. The sensitivity range was from 0.41 to 4.46 pg/mL; intra- and inter-assay coefficients of variation (CV) ranged from 2.9 to 3.1% and from 6.7 to 9.8%, respectively, for cell culture supernatants; for serum samples, the intra- and inter-assay CV ranged from 3.3 to 4.6%, and from 5.2 to 8.8%, respectively. Samples were assayed at least in triplicate. Quality control pools of low, normal, or high concentrations for all parameters were included in each assay (Human IP-10 Controls, R&D Systems Inc., Minneapolis, MN, USA).

Cell protein extraction and measurement to normalize chemokine secretion were performed as reported elsewhere [42]. Briefly, the obtained results were normalized by total cell protein amount. Protein extraction was performed in the 96-well plates using 1 M NaOH for 5 min, followed by incubation with Bradford reagent for 10–15 min. Protein concentration was measured by spectrophotometry at 595 nm.

### 4.5. RNA Extraction, Reverse Transcription, and Real-Time Quantitative Polymerase Chain Reaction (RT-qPCR)

Total RNA was extracted using TRIzol RNA Isolation Reagents (Invitrogen, Waltham, MA, USA), according to the manufacturer’s instructions. Single-stranded cDNA was obtained by reverse transcription of 500 ng-1 μg of total RNA and RT-qPCRs were performed using 7500 Real-Time System (Applied Biosystems, Waltham, MA, USA), using SYBR-green fluorophore and the manufacturer’s software (7500 Software v2.05) was used for fluorescence intensity analysis. Relative amounts were evaluated using the 2^−∆∆Ct^ method and normalized for the β-actin content. Data were expressed as fold increase compared with IFNγ and  TNFα, taken as 1. As follow, the primers used for CXCL10 were: forward (TTCCTGCAAGCCAATTTTGT), reverse (ATGGCCTTCGATTCTGGATT); for β-actin, forward (CTGAACCCCAAGGCCAAC), reverse (AGCCTGGATAGCAACGTACA).

### 4.6. Sodium Dodecyl Sulphate-PolyAcrylamide Gel Electrophoresis (SDS-PAGE) and Western Blot Analysis

Western blot analysis was performed in total protein extracts from cells, as previously described [21,58]. Hfaec and Hdf were homogenized in ice-cold lysis buffer (50 mM Tris–HCl, pH 7.5; 150 mM NaCl; 1 mM ethylenediaminetetraacetic acid, EDTA; 1% Triton; 0.25% SDS; 25 mM Na_3_VO_4_) supplemented with Complete Protease Inhibitor Mixture (Roche Diagnostics, Deutschland GmbH, Mannheim, Germany) and centrifuged for 15 min at 4 °C at 10,000× *g*. The supernatant was collected and protein concentration was measured using a Coomassie Bio-Rad protein assay kit (Bio-Rad Laboratories, Hercules, CA, USA). Protein aliquots were diluted in 4X reducing Laemmli’s sample buffer (Bio-Rad Laboratories, Hercules, CA, USA) and loaded onto 10% SDS-PAGE (using Prestained Precision Plus Protein Dual Color Standards, Bio-Rad Laboratories, Hercules, CA, USA). After SDS-PAGE, proteins were transferred to polyvinylidene difluoride membranes (PVDF; Amersham Hybond-ECL, GE Healthcare Italy, Milan, Italy). Membranes were blocked for 1 h at room temperature in TBS containing 5% BSA and 0.1% Tween 20 (TTBS) and washed in TTBS. The overnight incubation at 4 °C with primary antibodies diluted in TTBS, was followed by peroxidase-conjugated secondary IgG. The antibody dilutions were 1:1000 for phosphorylated (p-) and total STAT1, NF-kB, JNK, AKT 1/2/3, ERK 1/2, and 1:10,000 for peroxidase-conjugate secondary IgG. Membranes were stripped and reprobed with polyclonal anti-β-actin antibodies (dilution 1:1000) for protein content control.

Proteins were revealed by the enhanced chemiluminescence system (ECL plus, Amersham Biosciences, Little Chalfont, Buckinghamshire, UK). Image acquisition and densitometric analysis were performed by Image Quant Las 4000 software (GE Healthcare Italy, Milan, Italy) and Quantity One software (Bio-Rad Laboratories, Hercules, CA, USA).

### 4.7. Statistical Analysis

Statistical analysis was performed using GraphPad Prism 6 software (GraphPad Software, Inc., San Diego, CA, USA). Data were expressed as mean ± standard error (SE). The Kolmogorov-Smirnov test was used to verify the normal distribution of the data and parametric or non-parametric tests were used according to the variable’s distribution. Student’s *t*-test was used to compare different populations. A *p*-value < 0.05 was considered significant.

## 5. Conclusions

SSc implicates high mortality due to a combination of autoimmunity, vascular damage, and fibrosis. Given the complex disease progression involving skin and different organ complications, the therapeutic approach still represents a major critical challenge in clinics and basic research.

The finding that iloprost, in addition to the well-known vasodilator effect, can counteract CXCL10 release by human endothelial cells and fibroblasts, and associates with a significant lower chemokine blood level in SSc subjects, might open new scenarios for early treatments in SSc. Indeed, this chemokine is known to play a pivotal role at the onset of vascular and immune perturbation, likely acting as an early biomediator toward fibrosis, within the cross-talk among endothelium/fibroblasts/immune system. Interestingly, CXCR3^+^-binding chemokines, such as CXCL10, CXCL9, and CXCL11, are deeply involved in the pathogenesis of different Th1-driven skin diseases [59]. In this scenario, further studies are mandatory to deepen the biomolecular mechanism(s) involved in endothelium-dermal fibroblasts cross-talk mediated by chemokines.

Thus far, albeit this study is limited to human endothelial cells and fibroblasts without including, i.e., immunocytes, and is restricted to a small number of subjects, it could be hypothesis-generating for further studies, aimed to identify a more effective/safer and likely combined therapeutic regimen to early intervene in SSc disease.

## Figures and Tables

**Figure 1 ijms-23-10150-f001:**
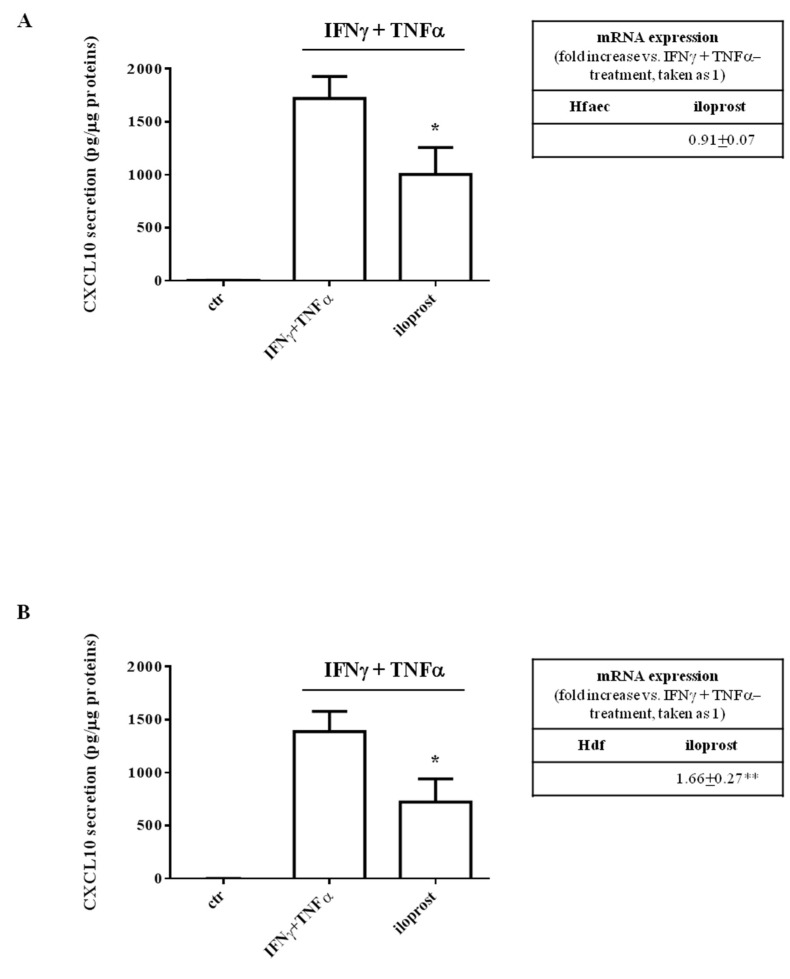
Iloprost-induced decrease of CXCL10 protein secretion in supernatants from cytokine-activated human endothelial cells (Hfaec) and dermal fibroblasts (Hdf). (**A**) In Hfaec, iloprost inhibited cytokine-induced CXCL10 release (* *p* < 0.05 vs. IFNγ and TNFα-induced secretion), without modifying chemokine-specific mRNA expression (inset of (**A**)). (**B**) In Hdf, iloprost reduced cytokine-induced CXCL10 release (* *p* < 0.05 vs. IFNγ and TNFα-induced secretion) and increased CXCL10 mRNA expression (inset of (**B**)) (** *p* < 0.01). Chemokine secretion is expressed as pg/µg of total proteins (mean ± SE). Data were obtained from five experiments with different cell preparations. mRNA expression is reported as a fold increase with respect to IFNγ and TNFα-induced expression, taken as 1. Data were obtained from three experiments with different cell preparations and reported as mean ± SE.

**Figure 2 ijms-23-10150-f002:**
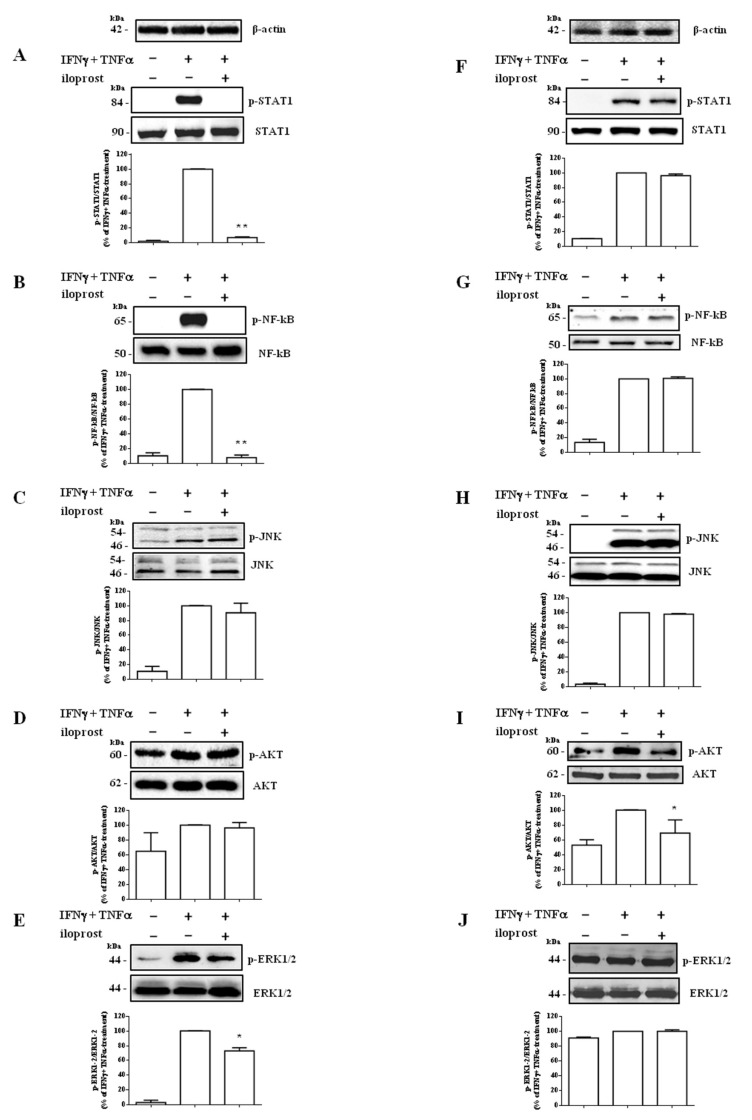
Iloprost-induced targeting of IFNγ and TNFα-dependent signaling cascade in human endothelial cells (Hfaec) and dermal fibroblasts (Hdf). (**A**–**E**) Western blot analysis in Hfaec showed that iloprost virtually prevented IFNγ and TNFα-induced activation of STAT1 and NF-kB (** *p* < 0.01 as from optical density/OD analysis) and significantly decreased cytokine-induced activation of ERK1/2 (* *p* < 0.05 as from OD analysis), without affecting JNK and AKT activation status. (**F**–**J**) Western blot analysis in Hdf showed that iloprost did not target cytokine-induced phosphorylation level of almost any tested path, except for AKT, which was reduced by the drug (* *p* < 0.05). Each panel depicts a representative blot over the diagram reporting OD, expressed as phosphorylated/total protein ratio (% of IFNγ and TNFα-induced treatment/activation, mean ± SE). β-actin was used as a sample loading control. Experiments were performed at least three times with five different cell preparations and reported as mean ± SE.

**Figure 3 ijms-23-10150-f003:**
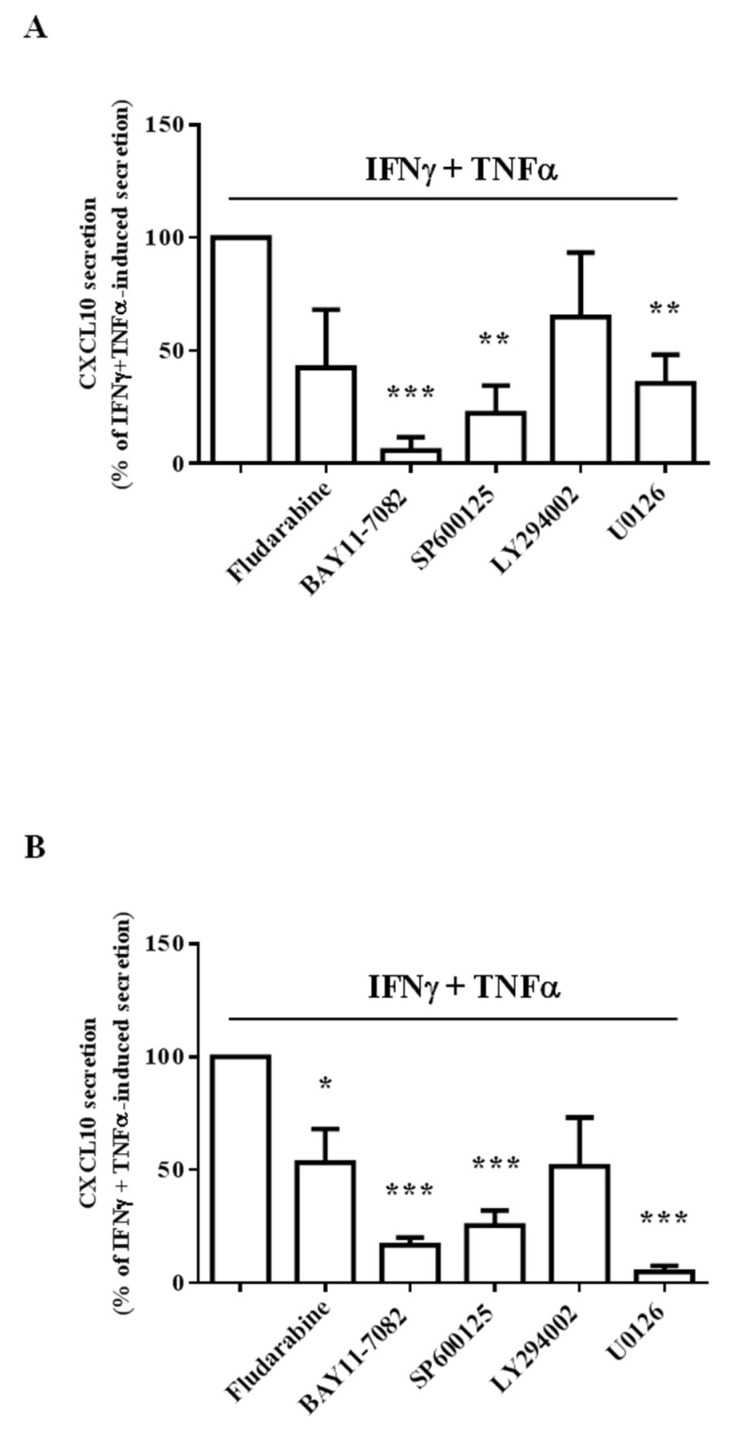
Different intracellular path engagement in CXCL10 protein release by human endothelial cells (Hfaec) and dermal fibroblasts (Hdf). (**A**) In Hfaec, cytokine-induced CXCL10 secretion (taken as 100%) was almost prevented by the specific NF-kB path inhibitor BAY11-7082 (*** *p* < 0.001) and significantly reduced blocking JNK and ERK1/2 (** *p* < 0.01) with SP600125 and U0126, respectively. The use of the specific inhibitors of STAT1 and AKT, fludarabine and LY294002 respectively, seemed not to significantly affect CXCL10 release. (**B**) The specific inhibition of ERK1/2 with U0126 almost nullified cytokine-induced CXCL10 release in Hdf (*** *p* < 0.001). NF-kB (BAY11-7082) and JNK (SP600125) blockage significantly decreased the chemokine (*** *p* < 0.001); to a lesser extent, STAT1 inhibition (fludarabine) also reduced CXCL10 secretion (* *p* < 0.05). The data were reported as mean ± SE.

**Figure 4 ijms-23-10150-f004:**
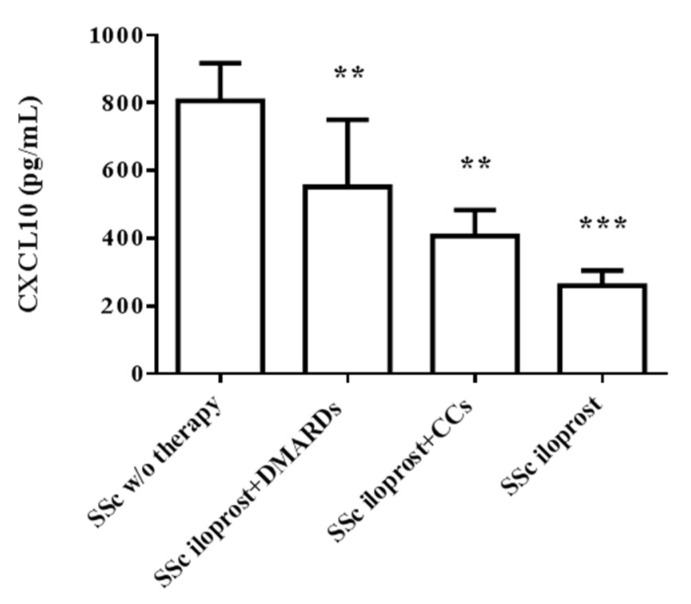
Circulating CXCL10 in SSc subjects taking iloprost (*n* = 25). SSc subjects taking iloprost either alone (*n* = 7) or in addition to DMARDs (*n* = 9) or CCs (*n* = 9) showed lower serum level of chemokine, as compared to the group of sex/age-matched SSc subjects without treatment (*n* = 20) (*** *p* < 0.001, ** *p* < 0.01). Data were derived from triplicate assays and were expressed as pg/mL (mean ± SE).

**Table 1 ijms-23-10150-t001:** Demographic, clinical, and therapeutic characteristics of SSc patients (*n* = 25) under iloprost therapy.

Characteristics	Value
Sex, female/male	2/23
Age, mean ± SD, years	60 ± 13.3
Disease duration, mean ± SD, years	19 ± 15.3
Form (limited SSc/diffuse SSc)	14/11
Raynaud Phenomenon (*n*/%)	25/100
Digital Ulcers (*n*/%)	14/56
Interstitial Lung Disease (*n*/%)	6/24
Pulmonary Arterial Hypertension (*n*/%)	1/4
Anti-nuclear Antibodies (ANA) (*n*/%)	25/100
Anti-topoisomerase I Antibodies (anti-Scl70) (*n*/%)	12/48
Anti-centromere Antibodies (ACA) (*n*/%)	8/32
**Drug therapy (in combination with iloprost)**	
**DMARDs (*n*/%)**	9/36
Hydroxychloroquine (*n*/%)	1/4
Azathioprine (*n*/%)	3/12
Cyclosporine (*n*/%)	3/12
Methotrexate (*n*/%)	2/8
**Corticosteroids (*n*/%)**	9/36
**Vasoactive drugs (*n*/%)**	25/100
Iloprost (single therapy) (*n*/%)	7/28

SD: Standard deviation; SSc: Systemic sclerosis; DMARDs: Disease-modifying anti-rheumatic drugs.

## Data Availability

All data are contained within the article. They are available on request from the corresponding authors (C.C. and V.R.) and can be accessed with a valid reason.

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
