# Peer review of "The Prostacyclin Analogue Iloprost Modulates CXCL10 in Systemic Sclerosis"

_ijms, 2022, doi:10.3390/ijms231710150_

Round 1
Reviewer 1 Report
Authors of manuscript titled ” The prostacyclin antalogue iloprost modulates CXCL10 in systemic sclerosis” provided and systematic approach to investigate if iloprost modulated CXCL10 in two in vitro cell lines and in subjects. Very specifically the authors demonstrated that iloprost inhibits CXCL10 secretion in vitro and clinically. The overall presentation or the results is clear but there are areas for improvements:
Minor
*Line 137, 205, 207, 219, 223 tense of sentence should be past.
*In vitro and vivo needs to be italicized
*Careful with the use of vs. in sentences. Line 34, 78, 87, 97, 157, 207, 332
*Line 192 “as shown by experiments with path inhibitors.” needs citations
*Having one sentence paragraphs makes the manuscript convoluted. Please reconsider rewording these areas.
*Where were the DMARD, Corticosteroids and Iloprost come from? What company?
*Figures have 1. mRNA data, 2. Blots, 3. CXCL10 release, 4. CSCL10 in patients, Methods have the order of ELISA assay (fig 3), WB (figure 3), RNA extraction etc (Fig. 1). I would consider rewriting the methods in the order that figures are presented and how they are described in the text.
*line 314 should be plural form and remove the ; and replace with and.
*line 329 remove ; and replace with and.
Major
*All figures are pixelated and this should be fixed.
*Figure 4. How many subjects per group? Please put in the legend.
*Statement “Our ongoing work includes studies to verify this hypothesis.” So why is this data not provided here in this study?
*Materials and Methods:
*please be specific in the “reagents for western blot# What is meant by this? Where are the molecular markers for SDS page listed and from where?
*Numerous places were there is not enough details that one could follow the authors’ methods so that the results could be reproduced.
*how were the cells lines characterized and maintained? With or without antibiotics
*ELISA assays: we don’t know which cells were used and what media was used and how much?
*IFNγ (1000 U/ml)+TNFα (10 ng/ml) The + sign needs to be written out and spacing
*Following sentence is really unclear as there is serum in medium as the authors report (0.1% BSA and vehicle). Also what is the vehicle here?
*Do the authors mean the the inhibitors in this sentence or what drugs? The drug concentration selected was maintained within the near-therapeutic range, ac-296 cording to the pharmacokinetic parameters (maximum drug concentration, Cmax, and 297 area under the time-concentration curve, AUC).
* Line 299 needs more information especially how the elisa samples were normalized and then assayed, volume of supernatants, centrifuged how long, temp, etc,
*What was the name of the commercially available kits exactly?
*How much sera was used in the commercially available kits? Was it diluted? If so in what buffer?
* Authors reported the following “Quality control pools of low, normal, or high concentrations for all parameters were included in each assay.” More exact information is need here and where is the data on these quality control pools.
* Cell protein extraction and measurement to normalize chemokine secretion were performed as reported elsewhere [28]. Authors really should consider elaborating these details and it is important that normalization is conducted before the ELISA is performed.
*This is also feedback for all western blot samples and analysis. How were the samples normalized before SDS-page to have WB conducted? It is important here in this study to make sure that normalization occurred before loading on the SDS gel. Traditionally gel loading controls are conduct. Were these conducted here? Why is the SDS method not described here. What percentage of the SDS gel were used to resolve the proteins?
*More clarity is needed in the description of the vehicle and the drug as defined by the authors (line 313 and earlier mentions)
*Spell out Abs Line 313
*Citation 30 provides two methods of how the cells were seeded and maintained. Please be specific here as what the authors did with both cell lines.
*Line “30 min in serum-free medium containing 0.1% BSA” actually has serum so it is not serum-free media or do the authors mean supplemented with 0.1% BSA?
Reviewer 2 Report
1. Writing style. The English is generally satisfactory but the text is often difficult to read because of the extensive use of abbreviations, some of which are inappropriately defined in the Abstract or in the Materials and Methods, not in the main text where they first appear.
1. Introduction and Abstract. The one-sentence paragraph mentioning iloprost as a prostacyclin analogue for treatment of scleroderma deserves at least some elaboration related to the preceding paragraph, specifically why the authors thought that iloprost might inhibit release or counter the effects of CXCL10. Strangely, the Abstract provides more information about the connection between these two molecules than does the Introduction. It is also unclear from the Abstract and Introduction whether the vasodilation agonist capacity of iloprost is in some way related to CXCL10 inhibition or whether the present observations are a novel finding which may or may not relate to vasodilation.
2. Results, Sec 2.1. The text is unclear. The reader is forced to search Methods to learn what Hfaec and Hdf stand for and that these are (presumably) primary cells. The text unnecessarily provides the raw data shown in Fig. 1 but requires its legend to know that the variances are SEM, not, more appropriately, standard deviations. The Method is absent any details as to how these in vitro studies were performed, only referencing two Reviews.
3. Materials and Methods. The sources of the chemicals and antibodies (Sec. 4.2) are incompletely described, sometimes just mentioning the state, not the city or the city, not the state, in the US where the company is located. The last sentence in Sec 4.2 does not correctly identify the company and where it is located. In addition, the first two paragraphs of Sec. 4.4 describe the cell culture method, not the assay for measuring CXCL10.
